# Effects of Dietary Supplementation of Chinese Yam Polysaccharide on Carcass Composition, Meat Quality, and Antioxidant Capacity in Broilers

**DOI:** 10.3390/ani13030503

**Published:** 2023-01-31

**Authors:** Yadi Chang, Jinzhou Zhang, Yan Jin, Jiahua Deng, Mingyan Shi, Zhiguo Miao

**Affiliations:** 1College of Animal Science and Veterinary Medicine, Henan Institute of Science and Technology, Xinxiang 453003, China; 2College of Life Science, Luoyang Normal University, Jiqing Road, Luoyang 471022, China

**Keywords:** Chinese yam polysaccharide, broiler, carcass performance, meat quality, antioxidant capacity

## Abstract

**Simple Summary:**

The purpose of this experiment was to investigate the effects of yam polysaccharide as a feed additive on the meat quality and antioxidant capacity of broilers. Our results showed that dietary yam polysaccharide could improve carcass performance, color difference, and shear force and enhance the antioxidant capacity of broilers.

**Abstract:**

The study aimed to evaluate the influences of the dietary supplementation of Chinese yam polysaccharide (CYP) on the carcass performance, antioxidant capacity, and meat quality of broilers. Three hundred and sixty healthy 1-day-old broilers with similar body weight (39 ± 1 g, gender balanced) were randomly divided into four groups (control, CYP1, CYP2, and CYP3 groups). In the control group, broilers were fed a basal diet with CYP, and the CYP1, CYP2, and CYP3 groups were fed diets supplemented with 250, 500, and 1000 mg/kg CYP, respectively. There were three replicates in each group, 30 birds in each replicate, and the feeding trial lasted for 48 days. Statistical analysis was performed using SPSS 17.0 (SPSS Inc., Chicago, IL, USA) by one-way analysis of variance. The results showed that compared with the control group, dietary supplementation with 500 mg/kg CYP can improve live weight, half-eviscerated carcass percentage, eviscerated carcass percentage, and thigh muscle percentage. Moreover, dietary supplementation with 500 mg/kg CYP can improve the contents of total antioxidant capacity (T-AOC), total superoxide dismutase (T-SOD), glutathione peroxidase (GPX), and glutathione s-transferase (GST) in serum (*p* < 0.05). Meanwhile, the mRNA expression levels of nuclear factor-erythroid 2-related factor 2 (*Nrf2*), heme oxygenase 1 (*HO-1*), quinone oxidoreductase (*NQO1*), superoxide dismutase 1 (*SOD1*), glutathione peroxidase 1 (*GPX1*), and catalase (*CAT*) in the liver; the mRNA expression levels of *HO-1*, *NQO1*, *GPX1*, and *CAT* in the breast muscle; and the mRNA expression levels of *NQO1*, *SOD1*, and *CAT* in the thigh muscle of broilers in the CYP2 group were significantly increased (*p* < 0.05). In addition, the yellowness and shear force of the thigh and breast muscles and the content of malondialdehyde (MDA) in the serum of broilers in the control group were higher than that in the CYP2 groups (*p* < 0.05). The results demonstrated that the CYP2 group had the best effect on improving meat quality. In conclusion, dietary supplementation with 500 mg/kg CYP can improve the meat quality of broilers by improving carcass quality, meat color, shear force, and antioxidant capacity.

## 1. Introduction

In recent years, because of the characteristics of fast growth, short feeding cycle, and high feed conversion rate of broilers, they have become the second largest breeding industry in China after pig breeding. In actual production, farmers usually adopt the mode of high-density broiler breeding to improve economic benefits. However, broilers are prone to stress in a higher-density environment, which damages intestinal health and affects the activity of antioxidant-related enzymes in serum, leads to the decline of immunity and antioxidant capacity of broilers, and thus reduces the performance and meat quality of broilers [1,2]. Then, breeding workers often improve these problems by adding low doses of antibiotics to the food. Due to the long-term abuse of antibiotics, not only is the microbiological balance of the animal intestinal tract severely damaged, but also the drug resistance of pathogenic strains is increasingly severe, threatening the survival of the whole organism, so various countries began to ban the use of feed antibiotics [3]. The use of natural, pollution-free, and residual-free plant extracts as additives instead of antibiotics to improve animal performance and meat quality has become a hot topic in current research.

Polysaccharides exist in the cell membranes of animals and plants and the cell walls of microorganisms (bacteria and fungi), which have a wide range of sources and have pharmacological effects in terms of antioxidation, blood pressure lowering, hypoglycemic, and antitumor [4,5,6]. Recently, a growing number of studies have shown that plant polysaccharides have significant effects on improving carcass composition, meat quality, and antioxidant activity [4,7]. Chinese yam is the dried rhizome of the perennial plant of the genus *Dioscorea*. It has a long history and extensive cultivation area, mainly originating in the warm areas of Africa and Asia, and then spreading all over the world. As a plant with medicinal and edible value, it has polysaccharides, protein, fatty acids, flavonoids, and other biologically active compounds [8,9]. Previous studies have shown that it has various functions, and it can be used for the majority of treatments of lung deficiency cough, spleen deficiency, chronic enteritis, chronic diarrhea, chronic gastritis, diabetes, enuresis, nocturnal emission, and underlying embolism in humans [10,11]. Previous studies have shown that sulfated Chinese yam polysaccharide (CYP) improves body weight, thymus index, and immune function in mice [12]. Furthermore, CYP has the functions of lowering blood glucose, reducing blood lipid, regulating the intestines and stomach, and improving antioxidant capacity [13,14,15]. Evidence showed that CYP had a strong scavenging ability in 2,2-diphenyl-1-picrylhydrazyl (DDPH), hydroxyl free radical, and superoxide free radical in vitro and could enhance the antioxidant activity of human endometrial epithelial cells, which may make it a potential candidate for a natural antioxidant [16]. Since previous studies of CYP have focused mainly on its free radical scavenging ability in vitro, this study aimed to research the effects of the dietary supplementation of CYP on the carcass composition, antioxidant capacity, and meat quality of broilers and provide a basis for scientifically adding CYP to the broiler diet.

## 2. Materials and Methods

### 2.1. Animal Care

In this study, all bird feeding programs were approved by the Animal Care and Use Committee of the Henan Institute of Science and Technology (No. 2020HIST018, Xinxiang, China).

### 2.2. Experimental Diets and Animal Management

Three hundred and sixty healthy 1-day broilers with similar body weight (39 ± 1 g, sex balance) were purchased from Henan Fengyuan Poultry Co., Ltd., Xinxiang, China, and the broilers were randomly divided into four groups: basal diet (CON group) and basal diet supplemented with 250, 500, 1000 mg/kg CYP (CYP1, CYP2, and CYP3 groups). Each group consisted of three replicates with 30 broilers per replicate. Additionally, data related to feed intake were prepared for publication as part of another article, and data related to average daily feed intake (g/d) are as follows: 1–28 d (control groups, 23.50 ± 0.75; CYP1 groups, 27.36 ± 0.23; CYP2 groups, 28.88 ± 0.03; and CYP3 groups, 26.06 ± 0.01), 29–48 d (control groups, 89.16 ± 1.38; CYP1 groups, 97.72 ± 0.02; CYP2 groups, 98.78 ± 0.09; and CYP3 groups, 97.19 ± 0.03), 1–48 d (control groups, 56.33 ± 0.50; CYP1 groups, 62.54 ± 0.14; CYP2 groups, 63.83 ± 0.02; and CYP3 groups, 61.62 ± 0.04). The nutritional requirements of broilers from the National Research Council were used to formulate the basal diet, and it was fed for 48 days [17]. Table 1 shows the composition and the levels of nutrients in the basal diet. Experimental broilers were reared in cages, and feed was available ad libitum. During the rearing period, all broilers were kept under constant light for 24 h, and room temperature was at 32 °C for the first 3 days and then gradually cooled to 26 °C on the 21st day. The crude CYP used in the experiment was purchased from Shaanxi Hana Biological Technology Co., Ltd., Xi’an, China, with a particle size pass 80 mesh, drying loss weight less than 5%, and polysaccharide content greater than or equal to 30%, and its monosaccharide types included glucose, 99.48%, and galactose, 0.52%).

### 2.3. Slaughter Performance Measurement

At 48 days of age, 24 broilers of similar live weights (6 broilers in each group, sex balanced) were randomly selected. After fasting for 12 h, they were weighed, and their venous blood from the wing was placed in centrifuge tubes without anticoagulant and allowed to condense overnight at 4 °C. Serum was collected after centrifugation (3000 g, 10 min, at 4 °C) and stored at −70 °C until analysis. Then, birds were slaughtered according to the procedures presented by Geldenhuys et al. [18]. After bloodletting, feather extraction and beak shell and foot skin removal, the carcass weight was taken. Viscera were removed; only the heart, liver, glandular stomach, and muscle stomach (excluding contents and keratin membrane) were retained, and abdominal fat was called half-eviscerated weight. The heart, liver, glandular stomach, muscular stomach, abdominal fat, head, and feet (the foot is divided from the hock joint, and the head is cut off from the link of the first cervical spine) were removed and then weighted to determine the dressing percentage. The thigh and breast muscles were taken out and weighed, and the percentages of the breast and thigh muscles were calculated. The meat samples were refrigerated at 4 °C for 24 h, and then the breast muscle and thigh muscle were divided into two parts. One was used to measure the meat color and shear force, while the other was frozen with liquid nitrogen and stored at −70 °C until analysis. The carcass-performance-related calculation formula is as follows:Dressing percentage (%) = (carcass weight/live weight) × 100%;
Half-eviscerated carcass percentage (%) = (half-eviscerated weight/live weight) ×100%;
Eviscerated carcass percentage (%) = (eviscerated weight/live weight) × 100%;
Breast muscle percentage (%) = (weight of breast muscle on both sides/live weight) × 100%;
Thigh muscle percentage (%) = (weight of thigh muscle on both sides/live weight) × 100%.

### 2.4. Color Measurement

After being stored at 4 °C for 24 h, the meat colors (including lightness, redness, and yellowness, which are L*, a*, and b* values) were determined by a colorimeter (Konica Minolta CR-410, Sensing Inc., Osaka, Japan). White tiles (L* 93.65, a* −0.88, and b* 3.24) were used as standard.

### 2.5. Shear Force Measurement

Shear force was measured with a Warner-Bratzler shear device (Zwick Roell Group, Ulm, Baden Wuerttemberg, Germany) in accordance with the method described by Liu et al. [19]. The thawed thigh and breast muscles were cut into three cubes of approximately 1 × 1 × 3 cm^3^ along the direction of the myofibrils. A shear instrument was used for longitudinal shear along the direction of the vertical myofibril, the shear force value was recorded, and the average value was calculated several times.

### 2.6. Serum Antioxidant Measurement

Serum was collected after centrifugation of blood samples clotted overnight and loaded into 1.5 mL of sterile tubes. Then the concentrations of glutathione peroxidase (GPX), total superoxide dismutase (T-SOD), glutathione s-transferase (GST), and malondialdehyde (MDA) in the serum of broilers were measured by the ELISA kit purchased from the Nanjing Jing Bioengineering Research Institute (Nanjing, China). The microassay was used to measure the total antioxidant capacity (T-AOC). The specific steps are as follows: preheat (30 min, adjust wavelength to 593 nm) → calibration → add sample to test tube, stand for 10 min → add sample (96-well plate) → measure OD value → calculate.

### 2.7. Quantitative Real-Time Polymerase Chain Reaction

Table 2 shows primer sequences designed according to the known sequences stored in GenBank. Gene expressions of nuclear factor-erythroid 2-related factor 2 (*Nrf2*), quinone oxidoreductase (*NQO1*), heme oxygenase 1 (*HO-1*), catalase (*CAT*), superoxide dismutase 1 (*SOD1*), and glutathione peroxidase 1 (*GPX1*) in liver and breast and thigh muscles were measured by quantitative real-time polymerase chain reaction (qRT-PCR). TRIzol reagent was used to extract total RNA from breast and thigh muscles and liver (Takara Bio Inc., Tokyo, Japan). The concentration and purity of the extracted RNA were measured using a spectrophotometer (Implen, Westlake Village, CA, USA) at an optical density of 1.8 ≤ 260/280 ≤ 2.2. The synthesis of complementary DNA (cDNA) obtained using the PrimeScript™ RT Reagent Kit (Takara Bio Inc., Tokyo, Japan). qRT-PCR was performed with the ViiA™ 7 Real-Time PCR System using a SYBR Green RT-PCR kit from Bio-Rad (Hercules, CA, USA). The mRNA expression levels were calculated by the 2^−ΔΔCT^ method.

### 2.8. Statistical Analysis

Statistical analysis was performed by one-way ANOVA using SPSS 17.0 (SPSS Inc., Chicago, IL, USA). Duncan’s multiple range tests were used for multiple comparison among groups. Data were expressed as mean ± SEM (standard error of means), and there was a significant difference at *p* < 0.05.

## 3. Result

### 3.1. Slaughter Performance

Table 3 shows the effects of the dietary addition of CYP on broilers’ slaughter performance. The liver weight, half-eviscerated carcass percentage, eviscerated carcass percentage, and thigh muscle percentage of broilers in the CYP2 group were higher than those in the control group (*p* < 0.05). Meanwhile, the CYP2 group had a higher eviscerated carcass percentage and thigh muscle percentage compared with the CYP3 group. No differences in live weight and half-eviscerated percentage were determined between the CYP1, CYP2, and CYP3 groups (*p* > 0.05). Furthermore, dressing percentage and breast muscle percentage were not different among all groups (*p* > 0.05).

### 3.2. Color and Shear Force

Table 4 shows the effects of the dietary addition of CYP on the meat quality of broilers. In our study, the values of L* and a* were not differences in the thigh and breast muscles of broilers (*p* > 0.05). The b* value of the CYP2 group was lower than that of the control group (*p <* 0.05), and the shear force of the thigh and breast muscles of the broilers in all experimental groups (CYP1, CYP2, and CYP3 groups) was lower than those in the control group (*p <* 0.05). Furthermore, the CYP1 and CYP2 groups had lower shear force of thigh muscle compared with the CYP3 group, and the CYP1 group had a lower b* value of the thigh muscle compared with the CYP2 and CYP3 groups, respectively (*p <* 0.05).

### 3.3. Antioxidant Enzyme Activities

As shown in Table 5, the activities of GST, GPX, and T-AOC in the serum of all experimental groups were higher than those of the control group (*p* < 0.01), while the activity of MDA was on the contrary (*p* < 0.01). Meanwhile, the activity of GST in the CYP3 group was higher than that in the CYP1 group and lower than that in the CYP2 group (*p <* 0.05). The activity of T-SOD in the control and CYP1 groups was significantly lower than that in the CYP2 and CYP3 groups (*p <* 0.05). Among all experimental groups, no differences in serum T-SOD, T-AOC, GPX, and MDA were determined (*p* > 0.05).

### 3.4. Gene Expression

As shown in Figure 1, Figure 2 and Figure 3, compared with the control group, in the liver, all experimental groups upregulated the *Nrf2*, *HO-1*, *NQO1*, and *SOD1* mRNA expression levels (*p <* 0.05), and the CYP2 group upregulated the *GPX1* and *CAT* mRNA expression levels (*p* < 0.05). In the breast muscle, all experimental groups upregulated the *HO-1* mRNA expression levels (*p <* 0.05), the CYP1 and CYP2 groups upregulated the *NQO1* mRNA expression levels (*p <* 0.05), and the CYP2 group upregulated the *CAT* and *GPX1* mRNA expression levels compared with the control group (*p <* 0.05). Meanwhile, there was no significant difference in *Nrf2* and *SOD1* mRNA expression (*p* > 0.05) among all groups (control, CYP1, CYP2, and CYP3 groups). In the thigh muscle, the CYP2 group upregulated the *HO-1*, *NQO1*, *SOD1*, and *CAT* mRNA expression levels compared with the control group (*p <* 0.05), while *Nrf2* and *GPX1* mRNA expressions were not significantly different among all groups (*p* > 0.05).

## 4. Discussion

A measure of differences in nutrient deposition in tissues is carcass traits [20]. Furthermore, the performance of meat production can be measured by slaughter and total carcass yield [9,21]. In our study, dietary CYP supplementation could significantly improve the live weight, half-eviscerated percentage, eviscerated carcass percentage, and thigh muscle percentage of broilers, and the group of 500 mg/kg CYP had the best effect. There is evidence that blueberry extract as a feed additive of natural plant extracts can improve the growth performance and carcass characteristics of broiler breast and thigh meat in the study of Ölmez et al. [22]. A previous study also found that the dressing, carcass, and breast and thigh weight of quail can be improved by dietary supplementation with clove oil (*Syzygium aromaticum*) [23]. Furthermore, the results of Miao et al. indicated that expanded chitosan significantly increased the slaughter rate and lean meat rate of growing-finishing pigs [24]. Our results are similar to previous studies, suggesting that the dietary supplementation of CYP may be used as a potential feed additive to improve the broilers’ carcass traits.

The appearance, texture, and other physiological characteristics of meat products directly affect consumers’ willingness to buy meat products. In a study of the New Zealand meat market, Moore et al. found that color value is considered an indicator of meat safety and quality [25], and it has been reported that a higher a* value and lower L* and b* values are more popular [26]. Shear force, an important indicator to evaluate meat tenderness, is one of the major attributes of sensation that determine whole acceptability, and the related study has shown that lower shear force and better meat quality generally reflect thinner muscle fibers and higher muscle moisture content [27,28]. We found that the dietary addition of 250 mg/kg CYP and 500 mg/kg CYP can reduce the b* value in the breast and thigh muscles of broilers at 48 days, but there is no difference in L* and a* values, which was opposite to the result of Wang et al. [29] and similar to that of Wang et al. [30]. Furthermore, our study showed that dietary supplementation with CYP resulted in lower shear force in broilers after slaughter, and the 500 mg/kg CYP group has the best result. A similar result was found in the effect of *Camellia oleifera* cake polysaccharides on meat quality in yellow broilers [29]. Therefore, it may be a way to improve the meat quality of broilers that add CYP to regulate the color and shear force of meat.

The antioxidant status is an important factor that affects host health, and the previous study showed that MDA is the primary degradation product of lipid peroxidation; its content is correlated with oxidative damage [31]. The opposite is that GPX and superoxide SOD are usually regarded as enzyme-free radical scavengers in cells and the important antioxidant enzyme that converts superoxide anion to hydrogen peroxide in cellular antioxidant reactions [32,33]. In addition, previous studies have suggested that CYP can improve oxidation stability. According to Zhu et al., CYP can effectively remove DPPH, ABTS+, and ·OH free radicals, and its scavenging ability on DPPH free radicals is equivalent to ascorbic acid [34]. Chen et al. successfully prepared yam oligosaccharides by hydrolyzing them with H_2_O_2_, and the results showed that yam oligosaccharides were a suitable hydroxyl radical scavenger [35]. Our study found that dietary addition with 500 mg/kg CYP can improve serum SOD and GPX activities and decrease the activity of serum MAD of broilers, thus improving the activity of serum T-AOC of broilers, which was similar to the study of Tan in 2014 [36], which found that the dietary supplementation of CYP could significantly improve the activity of GPX and decrease the content of serum MDA in yellow chickens. In addition, in poultry muscle, due to the lower fat content and high content of polyunsaturated fatty acids (PUFA), the unsaturated degree of the muscle membrane is increased. Therefore, poultry meat is easily oxidized, leading to a decrease in meat quality and nutritional value. In addition, it can reduce meat tenderness by reducing the activity of calcium-activating enzymes [37]. Lipid oxidation increases the permeability of cell membranes and decreases the water retention of muscles [38]. Thus, an antioxidant balance is essential for normal physiological and metabolic function in animals. *Nrf2*, a member of the basic leucine zipper transcription factor family, is a crucial transcription factor that maintains the balance of reductant–oxidant and signal transduction, which has a cellular protective effect on oxidative stress [39,40]. In addition, *Nrf2* can regulate *NQO1*, *CAT*, *GPX*, *SOD*, and a series of downstream antioxidant-related genes. *HO-1* is a target gene regulated by the *Nrf2* signaling pathway and is an antioxidant enzyme. *HO-1* was found to have antioxidant, anti-inflammatory, immunomodulatory, and antiapoptotic effects, and can decompose heme to produce biliverdin, free iron, and carbon monoxide (CO). Furthermore, reduced bilirubin and byproduct biliverdin can effectively remove activated oxygen to resist peroxide free radicals, hydroxyl groups, oxides, and peroxynitrite [41]. Ho-1 promoter activity can be directly regulated by *Nrf2*. In this study, in the 500 mg/Kg CYP group, *Nrf2* in the liver had a higher level of mRNA expression, and the expression levels of the *NQO1*, *HO-1*, *SOD1*, *GAX1*, and *CAT* genes downstream of *Nrf2* were also upregulated. Although the expression level of *Nrf2* in the breast muscle and the thigh muscle showed an upward trend, the difference was not significant; the expression levels of the downstream genes *HO-1*, *NQO1*, and *CAT* were significantly increased; and the expression levels of *GPX1* in the breast muscle and *SOD1* in the thigh muscle were also significantly increased. The results were similar to previous studies of natural polysaccharides in broilers [42,43]. Therefore, our results suggested that dietary supplementation with CYP can improve the antioxidant capacity of broilers by regulating the activity of serum antioxidant enzymes and the mRNA expression of related genes in the Nrf2 signaling pathway.

## 5. Conclusions

The dietary addition of yam polysaccharide had positive effects on the slaughter performance, serum antioxidant capacity, antioxidant-related genes in the liver and muscle, and meat quality of broilers. In addition, compared with 250 mg/kg and 100 mg/kg in the diet, the effect of 500 mg/kg is more prominent. These results indicate that dietary 500 mg/kg CYP can significantly improve the meat quality of broilers by changing slaughter performance, b* value, shear force, and antioxidant capacity.

## Figures and Tables

**Figure 1 animals-13-00503-f001:**
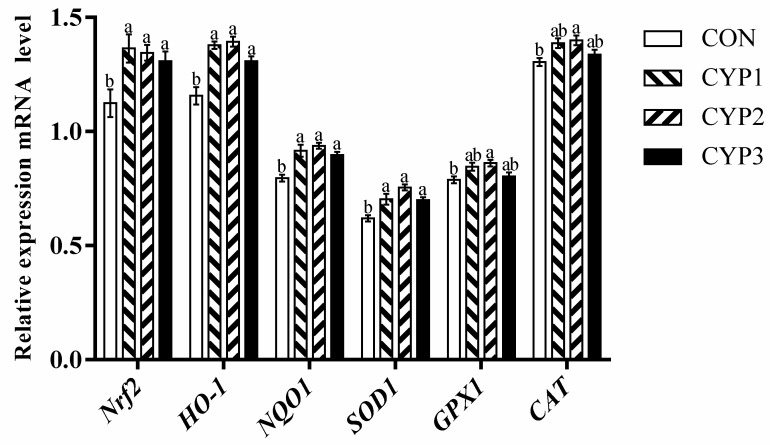
The *Nrf2*, *HO-1*, *NQO1*, *SOD1*, *GPX1*, and *CAT* mRNA expressions of liver in broilers. Different lowercase superscripts indicate significant differences (*p* < 0.05). CON = control diet; CYP 1 = control diet adds 250 mg CYP per kilogram; CYP 2 = control diet adds 500 mg CYP per kilogram; CYP 3 = control diet adds 1000 mg CYP per kilogram. *Nrf2*, nuclear factor-erythroid 2-related factor 2; *HO-1*, heme oxygenase 1; *NQO1*, quinone oxidoreductase; *SOD1*, superoxide dismutase 1; *GPX1*, glutathione peroxidase 1; *CAT*, catalase.

**Figure 2 animals-13-00503-f002:**
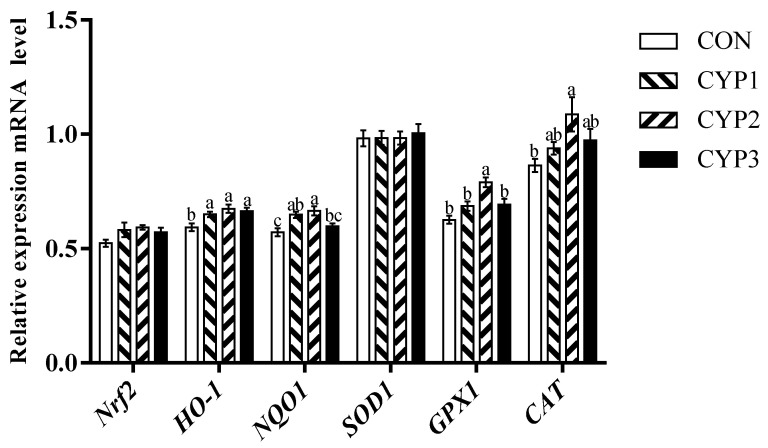
The *Nrf2*, *HO-1*, *NQO1*, *SOD1*, *GPX1*, and *CAT* mRNA expressions of breast muscle in broilers. Different lowercase superscripts indicate significant differences (*p* < 0.05). CON = control diet; CYP 1 = control diet adds 250 mg CYP per kilogram; CYP 2 = control diet adds 500 mg CYP per kilogram; CYP 3 = control diet adds 1000 mg CYP per kilogram. *Nrf2*, nuclear factor-erythroid 2-related factor 2; *HO-1*, heme oxygenase 1; *NQO1*, quinone oxidoreductase; *SOD1*, superoxide dismutase 1; *GPX1*, glutathione peroxidase 1; *CAT*, catalase.

**Figure 3 animals-13-00503-f003:**
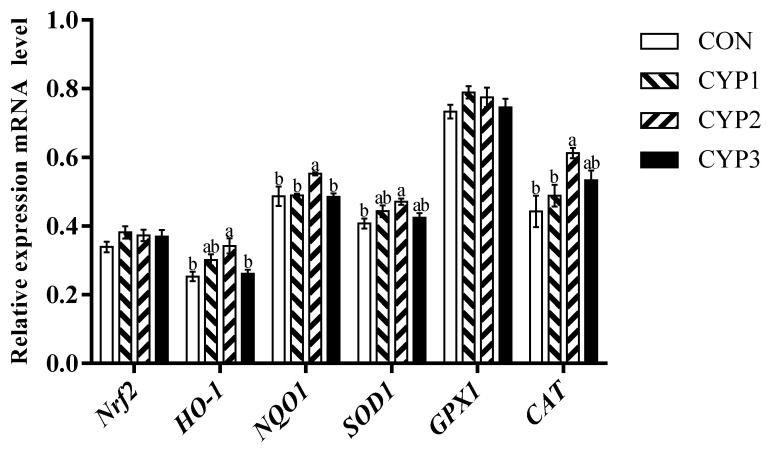
The *Nrf2*, *HO-1*, *NQO1*, *SOD1*, *GPX1*, and *CAT* mRNA expressions of thigh muscle in broilers. Different lowercase superscripts indicate significant differences (*p* < 0.05). CON = control diet; CYP 1 = control diet adds 250 mg CYP per kilogram; CYP 2 = control diet adds 500 mg CYP per kilogram; CYP 3 = control diet adds 1000 mg CYP per kilogram. *Nrf2*, nuclear factor-erythroid 2-related factor 2; *HO-1*, heme oxygenase 1; *NQO1*, quinone oxidoreductase; *SOD1*, superoxide dismutase 1; *GPX1*, glutathione peroxidase 1; *CAT*, catalase.

**Table 1 animals-13-00503-t001:** Ingredient and nutrient levels of the basal diet in each feeding phase for broilers.

Items	1–28 d	29–48 d
Ingredients (%)		
Corn	60.00	63.50
Soybean meal	32.00	29.00
Wheat bran	1.00	-
Soybean oil	1.00	2.00
Fish meal	2.00	1.60
CaHPO4	1.30	1.30
Limestone	1.40	1.30
NaCl	0.30	0.30
Premix ^1^	1.00	1.00
Total	100.00	100.00
Nutrient levels (%)		
Metabolic energy (MJ/kg) ^2^	12.13	12.55
Crude protein	21.00	20.00
Calcium	1.00	0.90
Total *p*	0.65	0.60
Available *p*	0.45	0.35
Lysine	0.50	0.38
Methionine	1.10	1.00

^1^ Premixes provided per kg of basic diet: VA 3000 IU, VD_3_ 500 IU, VE 10 IU, VK_3_ 0.5 mg, VB_6_ 3.5 mg, VB_1_ 3.8 mg, D-pantothenic acid 10 mg, folic acid 0.5 mg, biotin 0.15 mg, Fe 80 mg, Cu 8 mg, Zn 75 mg, Mn 60 mg, Se 0.15 mg; ^2^ Metabolic energy was a calculated value, and others were measured values.

**Table 2 animals-13-00503-t002:** Primer sequences for quantitative RT-PCR analysis.

Genes (Accession) ^1^	Primer Sequence	Length (bp)
*Nrf2* (MN416129)	F:5′-AACACACCAAAGAAAGACCCTCCTG -3′ R:5′- TTCACTGAACTGCTCCTTCGACATC-3′	207
*SOD1* (NM_205064)	F:5′- GGTCATCCACTTCCAGCAGCAG-3′ R:5′- AACGAGGTCCAGCATTTCCAGTTAG-3′	377
*CAT* (NM_001031215)	F:5′- CTCTCAGAAGCCAGATGCCTTGAC-3′ R:5′- CAGCAACAGTGGAGAACCGTATAGC-3′	293
*GPX1* (NM_001277853)	F:5′-GAAGTGCGAGGTGAACGGGAAG-3′ R:5′-TGCAGTTTGATGGTCTCGAAGTGG-3′	228
*NQO1* (NM_001277621)	F:5′-AAGATTGAAGCGGCTGACCTGATC-3′ R:5′-AGGCGGCTCTTCCATGTACTCAG-3′	365
*HO-1* (HM237181)	F:5′-GAGTCTCCAACGCCACCAAGTTC-3′ R:5′-TCCTGCTTGTCCTCTCACTGTCC-3′	273
*β-actin* (L08165)	F:5′-CATTGAACACGGTATTGTCACCAACTG-3′ R:5′-GTAACACCATCACCAGAGTCCATCAC-3′	270

^1^*Nrf2* = nuclear factor-erythroid 2-related factor 2; *SOD1* = superoxide dismutase 1; *CAT* = catalase; *GPX1* = glutathione peroxidase 1; *NQO1* = quinone oxidoreductase; *HO-1* = heme oxygenase 1; *β-actin* = beta-actin.

**Table 3 animals-13-00503-t003:** Effects of dietary CYP supplementation on the slaughter performance of broilers.

Item	CYP Level (mg/kg) ^1^				SEM ^2^	*p*-Value
	CON	CYP1	CYP2	CYP3		
Live weight (kg)	1.42 ^b^	1.60 ^a^	1.68 ^a^	1.58 ^a^	0.067	0.029
Dressing percentage (%)	89.96	91.02	92.20	92.62	0.838	0.087
Half-eviscerated carcass percentage (%)	79.70 ^b^	82.77 ^a^	82.87 ^a^	81.07 ^ab^	1.216	0.024
Eviscerated carcass percentage (%)	67.49 ^b^	69.56 ^ab^	70.40 ^a^	67.75 ^b^	1.315	0.024
Breast muscle percentage (%)	12.06	12.71	13.03	12.35	0.635	0.571
Thigh muscle percentage (%)	15.93 ^b^	17.89 ^a^	18.01 ^a^	16.50 ^b^	0.826	0.050

^1^ CON = control diet; CYP 1 = control diet adds 250 mg CYP per kilogram; CYP 2 = control diet adds 500 mg CYP per kilogram; CYP 3 = control diet adds 1000 mg CYP per kilogram; ^2^ SEM, standard error of the mean; ^a,b^ In the same line, same lowercase letters in the table indicate no significant difference (*p* > 0.05), but there are significant differences between different lowercase letters (*p <* 0.05).

**Table 4 animals-13-00503-t004:** Effects of dietary CYP supplementation on the meat quality of broilers.

Item ^2^	CYP Level (mg/kg) ^1^				SEM ^3^	*p*-Value
	CON	CYP1	CYP2	CYP3		
Breast muscle						
L*	52.50	52.08	52.08	52.20	0.426	0.738
a*	4.94	5.02	5.22	4.96	0.182	0.416
b*	11.29 ^a^	10.46 ^b^	10.46 ^b^	10.56 ^a^	0.288	0.039
Shear force (N)	16.31 ^a^	15.48 ^b^	14.71 ^b^	14.88 ^b^	0.270	0.002
Thigh muscle						
L*	55.96	55.44	55.24	55.31	0.481	0.481
a*	10.22	10.04	9.71	10.22	0.201	0.084
b*	13.88 ^a^	12.57 ^ab^	11.90 ^b^	12.94 ^ab^	0.572	0.047
Shear force (N)	27.48 ^a^	21.94 ^c^	21.83 ^c^	23.18 ^b^	0.533	0.001

^1^ CON = control diet; CYP 1 = control diet adds 250 mg CYP per kilogram; CYP 2 = control diet adds 500 mg CYP per kilogram; CYP 3 = control diet adds 1000 mg CYP per kilogram; ^2^ L* = lightness; a* = redness; b* = yellowness; ^3^ SEM = standard error of the mean; ^a–c^ In the same line, same lowercase letters in the table indicate no significant difference (*p* > 0.05), but there are significant differences between different lowercase letters (*p <* 0.05).

**Table 5 animals-13-00503-t005:** Effect of dietary CYP supplementation on the serum antioxidant of broilers.

Item ^2^	CYP Level (mg/kg) ^1^				SEM ^3^	*p*-Value
	CON	CYP1	CYP2	CYP3		
T-SOD (pg/mL)	44.07 ^b^	43.96 ^b^	48.04 ^a^	47.49 ^a^	0.527	0.001
T-AOC (U/mL)	5.58 ^b^	5.89 ^a^	5.94 ^a^	5.88 ^a^	0.088	0.013
GPX (pmol/mL)	15.22 ^b^	16.50 ^a^	16.63 ^a^	16.57 ^a^	0.217	0.001
GST (ng/L)	459.74 ^d^	482.87 ^c^	530.34 ^a^	502.82 ^b^	4.693	0.001
MDA (nmol/L)	15.88 ^a^	15.40 ^b^	15.11 ^b^	15.27 ^b^	0.130	0.002

^1^ CON = control diet; CYP 1 = control diet adds 250 mg CYP per kilogram; CYP 2 = control diet adds 500 mg CYP per kilogram; CYP 3 = control diet adds 1000 mg CYP per kilogram; ^2^ T-SOD = total superoxide dismutase; T-AOC = total antioxidant capacity; GPX = glutathione peroxidase; GST = glutathione s-transferase; MDA = malondialdehyde; ^3^ SEM, standard error of the mean; ^a–d^ In the same line, same lowercase letters in the table indicate no significant difference (*p* > 0.05), but there are significant differences between different lowercase letters (*p <* 0.05).

## Data Availability

Data supporting the results of this study can be provided by the corresponding authors on reasonable demand.

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
