# Peer review of "Effects of Dietary Supplementation of Chinese Yam Polysaccharide on Carcass Composition, Meat Quality, and Antioxidant Capacity in Broilers"

_animals, 2023, doi:10.3390/ani13030503_

Round 1
Reviewer 1 Report
In the interesting manuscript Chang et al. examines how Chinese yam polysaccharide affect carcass composition, meat quality and antioxidant capacity in broilers. The experiments are well designed. However, the quality of the experiments and tables needs to be strongly improved.
Major comments:
1. Line 105: What does the particle size greater than 95.00% mean?
2. Line139: Please briefly describe the steps used to measure shear forces.
3. Line 147: Briefly explain the micro-assay method.
4. Line 147: No single assay can represent the total antioxidant capacity. Three different and complementary assay methods are recommended to evaluate the antioxidant activities of extract: total antioxidant capacity, the reducing power, and DPPH free radical scavenging.
5. Line175: Data on liver weights are not available in the table 3.
6. Table 3-table 6: Please explain which significance of difference between the two groups is represented by the p-values in the table.
7. Section 3.3: The activity of GST should be measured rather than the concentration.
8. Line 203: The concentrations of T-AOC in serum? Again, the activity of T-AOC should be measured.
9. Section 3.4: Gene expression data should be represented by a histogram.
10. Gene expression analysis should be also strengthened by western-blot analysis
Author Response
Dear Reviewer,
Thank you very much for your questions. All questions from you we have answered as follows:
- Line 105: What does the particle size greater than 95.00% mean?
Response 1: Thank you very much for your advice. Particle size greater than 95.00% refers to polysaccharide content, which we have modified. (In red: Line 107).
- Line139: Please briefly describe the steps used to measure shear forces.
Response 2: Thank you very much for your advice. We have made corresponding additions to the shear force measurement method. (In red: Line 143-145).
- Line 147: Briefly explain the micro-assay method.
Response 3: Thank you very much for your advice. We have supplemented the method of microanalysis accordingly. (In red: Line 152-155).
- Line 147: No single assay can represent the total antioxidant capacity. Three different and complementary assay methods are recommended to evaluate the antioxidant activities of extract: total antioxidant capacity, the reducing power, and DPPH free radical scavenging.
Response 4: Thank you very much for your advice. Your proposal is very good, but due to sample and expense, we haven't done relevant experiments. We will consider these in the next step.
- Line175: Data on liver weights are not available in the table 3.
Response 5: Thank you very much for your advice. It is live weight and we have modified it.
- Table 3-table 6: Please explain which significance of difference between the two groups is represented by the p-values in the table.
Response 6: Thank you very much for your advice. The p value represents the significant relationship between the two groups and is annotated below Table 3 - table6. (In red: Line 191-193, 206-208, and 221-223).
- Section 3.3: The activity of GST should be measured rather than the concentration.
Response 7: Thank you very much for your advice. We have modified concentration to activity. (In red: Line 210).
- Line 203: The concentrations of T-AOC in serum? Again, the activity of T-AOC should be measured.
Response 8: Thank you very much for your advice. We have modified concentration to activity. (In red: Line 212).
- Section 3.4: Gene expression data should be represented by a histogram.
Response 1: Thank you very much for your advice. Gene expression data as well as modified histograms.
- Gene expression analysis should be also strengthened by western-blot analysis
Response 1: Thank you very much for your advice. Your proposal is very good, but due to sample and expense, we haven't done relevant experiments. We will consider these in the next step.
In addition:
we have also used red font to show the parts that have been modified other than the problem.
Sincerely,
Zhiguo Miao

Reviewer 2 Report
General - please add statistical methods briefly to the abstract. Also, you report P<0.05 in two sentences of results in the abstract, but not two others. Please do all for consistency or can specify the statistical cut off in the statistical methods sentence in the abstract one time instead.L12 - recommend removing this first sentence because "green and safe" do not have definitive definitions. Would just include "as a feed additive" after the words "yam polysaccharide in the second sentence in L13.
L22 - Add "There were" before "three replicates..."
L49 - please elaborate on your definition of "antioxidant performance" here (and also maybe "immune performance"). what are some indicators of these that the references you site use? L53 - minor edit, remove "is" before "severely" L55 - please include your definition of "green" if using the term here L64 - Dioscorea should be italic since it is a genus L65 - change to "polysaccharides (e.g. starch)" since starch is a polysaccharides so they should be separate list items L68-70 - What species do these studies refer to? L77 - minor edit, change "be" to "make it" L78 - change " the studies of CYP" to "previous studies of CYP have" L91-92 - please change this to a complete sentence L98-99 - change " Based on the nutritional requirements of broilers from the National Research Council to formulate the basal diet and fed for 48 days " to " The nutritional requirements of broilers from the National Research Council were used to formulate the basal diet and it was fed for 48 days " L107 - did not see it mentioned what ingredient was lessened to add in the incremental levels of CYP, please add this statement if it is not in the article L114 - do you mean live weights? not liver weights? please correct if needed L126- breast and thigh percentage of what total weight? live? other? please specify Tables 3-6 - some inconsistencies in bold, underline, etc. formatting in these, please double check and correct unless these were intentional L242-248 - please explain the connection of these other additive types to CYP (are they all antioxidants?) or otherwise change to prior literature to compare with that fed CYP L251-263 - think two good additions here would be: 1) what region of the world/country prefers those color profile? because that can differ by regional preference. Or is there a study that found that there is a world-wide color preference overall? and 2) a quick explanation of whether shear force increase or decrease is preferable for tenderness for those reading where this is not their expertise L284 - you have the word "increases" twice in a row, please remove one of the words L309-311 - this conclusion is far too short for all of the measurement parameters covered and is too generalized to firmly conclude that CYP is superior. please revise the conclusion to cover all major categories of results and be more specific to the finding. Similar modifications to the abstract may be needed. In both the conclusions and abstract, it is not clear from the data that CYP2 treatment was superior in all cases, so this general statement should be revised to better represent the various findings throughout.Author Response
Dear Reviewer,
Thank you very much for your questions. All questions from you we have answered as follows:
Point 1: L49 - please elaborate on your definition of "antioxidant performance" here (and also maybe "immune performance"). what are some indicators of these that the references you site use?
Response 1: Thank you very much for your advice. Antioxidant performance and immune performance refer to the immunity and antioxidant capacity of broilers, respectively. The susceptibility of broilers to necrotizing enteritis and the overproduction of free radicals stimulated by heat stress were cited respectively. (In red: Line 47-49).
Point 2: L53 - minor edit, remove "is" before "severely"
Response 2: Thank you very much for your advice. We have changed the language. (In red: Line 52).
Point 3: L55 - please include your definition of "green" if using the term here
Response 3: Thank you very much for your advice. Green means natural and we have made modifications accordingly. (In red: Line 55).
Point 4: L64 - Dioscorea should be italic since it is a genus
Response 4: Thank you very much for your advice. We have changed the scientific name to italic format. (In red: Line 64)
Point 5: L65 - change to "polysaccharides (e.g. starch)" since starch is a polysaccharides so they should be separate list items
Response 5: Thank you very much for your advice. We have removed the‘starch. (In red: Line 67)
Point 6: L68-70 - What species do these studies refer to?
Response 6: Thank you very much for your advice. The species do these studies refer to humans. (In red: Line 71)
Point 7: L77 - minor edit, change "be" to "make it"
Response 7: Thank you very much for your advice. We have changed the language. (In red: Line 78).
Point 8: L78 - change " the studies of CYP" to "previous studies of CYP have"
Response 8: Thank you very much for your advice. We have changed the language. (In red: Line 79).
Point 9: L91-92 - please change this to a complete sentence
Response 9: Thank you very much for your advice. We have changed the language. (In red: Line 90, 91, and 93).
Point 10: L98-99 - change " Based on the nutritional requirements of broilers from the National Research Council to formulate the basal diet and fed for 48 days " to " The nutritional requirements of broilers from the National Research Council were used to formulate the basal diet and it was fed for 48 days "
Response 10: Thank you very much for your advice. We have changed the language. (In red: Line 100-101).
Point 11: L107 - did not see it mentioned what ingredient was lessened to add in the incremental levels of CYP, please add this statement if it is not in the article
Response 11: Thank you very much for your advice. In our experiment, yam polysaccharide was added to the basal diet without the change of nutrient composition in the basal diet, as mentioned in lines L19-L20 and L89-90 of the paper.
Point 12: L114 - do you mean live weights? not liver weights? please correct if needed
Response 12: Thank you very much for your advice. It is live weight and we have corrected it. (In red: Line 116).
Point 13: L126- breast and thigh percentage of what total weight? live? other? please specify Tables 3-6 - some inconsistencies in bold, underline, etc. formatting in these, please double check and correct unless these were intentional
Response 13: Thank you very much for your advice. The proportion of breast muscle and thigh muscle is the percentage of live body weight. The calculation method is explained in the formula after Line 130 in the paper. Table 3-6 has been checked and rectified.
Point 14: L242-248 - please explain the connection of these other additive types to CYP (are they all antioxidants?) or otherwise change to prior literature to compare with that fed CYP
Response 14: Thank you very much for your advice. These additive types are natural plant extracts with CYP. (In red: Line 264-266).
Point 15: L251-263 - think two good additions here would be: 1) what region of the world/country prefers those color profile? because that can differ by regional preference. Or is there a study that found that there is a world-wide color preference overall? and 2) a quick explanation of whether shear force increase or decrease is preferable for tenderness for those reading where this is not their expertise
Response 15: Thank you very much for your advice. We have made corresponding additions to the meat color investigation country and the relationship between shear force and meat quality. (In red: Line 274, 275, 279, and 280).
Point 16: L284 - you have the word "increases" twice in a row, please remove one of the words
Response 16: Thank you very much for your advice. We have made changes to the language. (In red: Line 300-303).
Point 17: L309-311 - this conclusion is far too short for all of the measurement parameters covered and is too generalized to firmly conclude that CYP is superior. please revise the conclusion to cover all major categories of results and be more specific to the finding. Similar modifications to the abstract may be needed. In both the conclusions and abstract, it is not clear from the data that CYP2 treatment was superior in all cases, so this general statement should be revised to better represent the various findings throughout.
Response 1: Thank you very much for your advice. We have supplemented and modified the conclusion accordingly. (In red: Line 335-338).
In addition:
we have also used red font to show the parts that have been modified other than the problem.
Sincerely,
Zhiguo Miao

Round 2
Reviewer 1 Report
Thank you very much~